# Multilevel analysis of factors associated with abortion among adolescents in Uganda insights from UDHS 2022 dataset

Stephen Mungau[1]*, Joan Nanteza[1,2], Genevieve Dupuis[3]

1 Department of Epidemiology and Biostatistics, School of Public Health, Makerere University, Kampala, Uganda, 2 Bugiri District Local Government, Bugiri, Uganda, 3 Department of Biostatistics, Boston University School of Public Health, Boston, Massachusetts, United States of America

* stevemungau8@gmail.com

## Abstract

Unsafe abortion is a major reproductive health challenge, causing 7.9% of global maternal deaths and 9.6% in East Africa. In Uganda, about 8% of maternal deaths result from unsafe abortions. Early sexual activity, poor access to sex education, restrictive laws and stigma push adolescents into unsafe practices. Limited safe services force many to use dangerous methods leading to severe complications and high maternal mortality. This study examined determinants and prevalence of abortion among Ugandan female adolescents using the 2022 Uganda Demographic and Health Survey dataset of 5,125 females aged 15–24 who had ever engaged in sexual activity. The dependent variable was binary (1 for ever terminated, 0 for never). Weighted data were analyzed using descriptive statistics, ordinary and mixed effect logistic regression models to explore individual- and cluster-level influences. Intra-class correlation and likelihood ratio tests assessed cluster variation. Findings showed 562 adolescents had ever aborted. Those whose first sex was before age 15 were 3.44 times more likely to abort compared to those aged 20–24 while those aged 15–17 were 2.24 times more likely. Married adolescents had twice the odds compared to never married, and cohabiting adolescents were 2.44 times more likely. Compared to those with education beyond secondary, adolescents with no education, primary and secondary schooling were 5.8, 2.99 and 3.01 times more likely to abort. Regional variations accounted for 16.8% of variance, with intra-class correlation of 4.9%. Overall, 11.0% of Ugandan female adolescents reported abortion. Key determinants included age at first sex, marital status, education, contraceptive use and internet use. Region-level factors contributed 4.9% of variation highlighting the need for cluster-level interventions alongside individual approaches.

the Creative Commons CC0 public domain dedication.

**Data availability statement:** The dataset used in the analysis can be accessed via the link https://github.com/stevemungau/Udhs_2022_dataset.git.

**Funding:** This study did not receive any specific grant or funding from any source be it commercial or public agency. However, SM and GD were supported by Hope4Kids organization under sponsorship number H4KI/BUN/1247A and the National Institutes of Health, Award Number T32GM140972 respectively for their graduate training and the funders had no role in study design, data collection and analysis, decision to publish or preparation of the manuscript.

**Competing interests:** The authors have declared that no competing interests exist.

## Introduction

Globally, an estimated 7.9 million unintended pregnancies occur among adolescents annually, with half ending in unsafe abortions [1]. According to the World Health Organization, an unsafe abortion is a procedure for terminating a pregnancy carried out either by individuals lacking the necessary skills or in an environment that does not conform to minimal medical standards, or both [2]. This phenomenon is particularly concerning in developing countries like Uganda, where access to safe and legal abortion services is often limited leading to high rates of unsafe abortion and associated risks to adolescent health and well-being [3]. In 2023, research findings indicated that approximately 10% of maternal deaths in Uganda were attributed to unsafe abortions [4].

Abortion among adolescents represents a critical public health issue in Uganda, bearing profound social, health and economic implications for both individuals and society at large [5]. An estimated 1 in 4 Ugandan women older than 19 experiences a pregnancy while were 15–19 years [6]. In Uganda, where adolescent fertility remains high [6,7], understanding the factors associated with abortion among this population is crucial for informing effective prevention and intervention strategies on abortion. This study explores this critical issue by employing a multilevel analysis approach to explore the individual, household and region-level determinants of abortion among adolescents in Uganda using data from the 2022 Uganda Demographic and Health Survey (UDHS).

## Background

Adolescence, characterized by rapid physical and psychological development poses unique challenges for sexual and reproductive health [8]. In Uganda, early sexual debut and high rates of unprotected sex among adolescents contribute to a significant burden of unintended pregnancies [9]. This burden is further amplified by limited access to modern contraceptive methods, particularly long-acting reversible contraception, limited access to comprehensive sex education, restrictive abortion laws and sociocultural stigma surrounding premarital sex and abortion [9,10]. Consequently, a substantial number of Ugandan adolescents resort to abortion often under unsafe conditions [11].

Unsafe abortions carried out by untrained providers or using dangerous methods often result in severe health complications and even death [12]. In Uganda, the prevalence of unsafe abortions among adolescents is a significant contributor to maternal mortality, prompting serious concerns regarding the reproductive health and long-term well-being of young girls [13,14]. This emphasizes the critical need for comprehensive reproductive health education, access to contraception and safe abortion services to protect the health and rights of adolescent girls and reduce maternal mortality rates [15]. In addition, unsafe terminations can lead to future fertility complications, including infection-related infertility, chronic pelvic pain and adverse pregnancy outcomes in later life [16] Beyond medical consequences, unwanted pregnancies and subsequent terminations can disrupt adolescents' education, limit their economic opportunities and negatively impact their mental and emotional health [17].

Adolescent abortion, encompassing induced abortion and miscarriages remains a complex and sensitive issue globally and pose significant challenges, particularly in low- and middle-income countries like Uganda [15]. In Uganda, where cultural, societal and legal stances on abortion are restrictive, adolescent abortion often occurs in secret and unsafe conditions, leading to increased health risks and consequences for young girls [18].

The factors associated with abortions among Ugandan adolescents are varied and interconnected [19,20]. Socioeconomic disparities, limited access to comprehensive sexual and reproductive health education, inadequate healthcare facilities, cultural norms and restrictive abortion laws collectively shape the experiences and decisions surrounding abortion among this demographic [18]. Furthermore, the geographical variability within Uganda adds another layer of complexity as regional disparities in healthcare infrastructure and socio-cultural norms may significantly impact adolescents' choices regarding pregnancy continuation or termination [21].

Recent studies have highlighted the importance of comprehensive approaches that take into account both individual-level determinants and contextual factors that may be influencing adolescent pregnancies and their outcomes [22]. However, comprehensive multilevel analyses integrating individual, household and region-level variables in the Ugandan context remain scarce. This study aims to bridge this gap by using advanced statistical methods to analyze the UDHS 2022 dataset and explore the hierarchical nature of factors associated with abortions among Ugandan adolescents.

The rich and nationally representative UDHS dataset offers a unique opportunity to study the complex determinants of abortions among adolescents in Uganda [6]. By utilizing a multilevel analytical framework, this research aims to provide evidence-based insights that can inform targeted interventions, policy formulations and healthcare strategies for mitigating the adverse outcomes associated with adolescent pregnancies and terminations.

## Methodology

### Study population and data source

This study used data from Uganda DHS data collected in 2022. The Uganda DHS is a national representative cross-sectional survey. The survey is carried out approximately every five years by the Uganda Bureau of Statistics (UBOS) in collaboration with the Ministry of Health (MoH) and with support from partners including the United Nations Population Fund (UNFPA), the United Nations Children's Fund (UNICEF), United Nations High Commission for Refugees (UNHCR) the Government of Uganda and other partners [23]. The 2022 UDHS survey employed a two-stage sampling method to establish a national representative sample. In the initial stage, 697 Enumeration Areas (EAs) were chosen, comprising 233 in urban areas and 464 in rural areas. The second stage was the selection of households from each EAs. In total, 20,631 non-refugee households and 2,276 refugee households were chosen from these EAs. From selected households, women aged 15–49 were eligible to be individually interviewed and in one-third of households, all men 15 – 54 were also eligible to participate in the survey [23]. For this study, 5,125 female adolescents between the age of 15–24 years who reported ever having sex were included and the dependent variable in this study was whether there was an abortion or not within the five years preceding the survey. The independent variables were chosen based on the existing literature as the most suitable representatives of the factors influencing abortion practices in Uganda. These variables included demographic characteristics (age, education, marital status), socio-economic factors (employment status, wealth index), reproductive health measures (history of contraceptive use, prior pregnancies, parity), and socio-cultural influences (religion, community norms around premarital sex and abortion).

### Statistical analysis

Stata 14 was used for data extraction, recoding and analysis. Weighting was done to ensure the accuracy and reliability of the findings using the corresponding weights at the individual and household that take into account the survey's design. This crucial step was undertaken prior to conducting any statistical analyses to ensure the data's representativeness and produce dependable estimates and standard errors.

Descriptive statistics were used to analyze categorical variables by calculating frequencies and percentages. To identify multicollinearity among independent variables, both correlation matrices and variance inflation factors (VIFs) were produced. Variables with a correlation coefficient of (>=0.4) were initially regarded as highly correlated. However, the final decision on retaining correlated variables was based on a combination of statistical significance and biological plausibility. For instance, if two variables were highly correlated but represented distinct constructs, both variables were retained in the model. The consideration of VIFs helped assess the severity of multicollinearity beyond simple correlation analysis.

The presence of clustering was assessed using the Intraclass Correlation Coefficient (ICC), while the Likelihood Ratio Test (LRT) and mixed-effect logistic regression were used to analyze factors associated with abortion at the cluster level. Variables with a p-value below 0.1 in bivariate analysis were considered for inclusion in the multivariable analysis. In the multivariable analysis, variables with a p-value below 0.05 were considered significantly associated with abortion among adolescents.

### Multilevel model

Multilevel analysis emerges as a robust methodology for understanding complex public health issues, including reproductive health outcomes. It allows for the simultaneous examination of individual-level factors (such as age, education and access to healthcare) and contextual determinants (such as regional norms, socioeconomic status and geographical variations) that influence health outcomes [24]. For instance, higher levels of education often correlate with improved knowledge about reproductive health, while increased unmet need for contraception at the regional level is associated with higher likelihood of unplanned pregnancies among young girls [15].

## Results

### Adolescent background characteristics

The study analyzed data from 5,125 sexually active adolescents aged 15–24 years using the 2022 Uganda Demographic and Health Survey (UDHS). Overall, 562 adolescents (11.0%) reported having ever terminated a pregnancy.

Finding in Table 1 revealed that the age at sexual debut plays a crucial role in the likelihood of abortion. Adolescents who initiated sexual activity at younger ages exhibited higher abortion rates: 15.9% among those who started before age 15, 11.3% for the 15–17 age group and notably decreased rates as the age at first sex increased, reaching 7.0% for those who began sexual activity after age 20.

Marital status significantly influenced abortion rates among adolescents. Cohabiting adolescents reported the highest rate 15.2%, followed by legally married adolescents 12.4% and 7.0% among those who had never married.

Socioeconomic factors such as education also significantly influence abortion rates. Adolescents with no education or primary education had higher abortion rates 12.2% and 11.5% respectively compared to those with tertiary or higher education, who had a noticeably lower rate of just 6.6%.

The study also found slight variation in abortion prevalence by place of residence. Adolescents living in urban areas (11.9%) were marginally more likely to terminate a pregnancy compared to their counterparts in rural areas (10.5%).

Fig 1 shows significant regional variation in abortion prevalence in Uganda, ranging from 6.4% in Teso to 14.9% in Acholi. High prevalence is also observed in Busoga (13.8%) and Bukedi (12.0%), while Kigezi (8.3%) and Karamoja (8.5%) report comparatively low levels. Most regions, including Kampala, Bugisu, West Nile and Bunyoro, cluster around 11%, highlighting the need for region-specific reproductive health interventions rather than uniform national approaches

### Assessing for variation in abortion at cluster level

As shown in Table 2, the intra-cluster correlation coefficient (ICC) for the regional random intercept model indicated that 4.9% of the variation in abortion outcomes could be attributed to differences between regions with a variance of 16.8%.

**Table 1. Distribution of ever having had an abortion, by selected background characteristics.**

| Factor | Proportion of adolescent abortion | | | Chi-square | p-value |
|---|---|---|---|---|---|
| | No (4,563) | Yes (562) | % of abortion | | |
| **Age at first sex** | | | | 45.91 | <0.001 |
| <15 | 766 | 145 | 15.9 | | |
| 15-17 | 2,547 | 324 | 11.3 | | |
| 18-20 | 1,130 | 86 | 7.1 | | |
| >20 | 120 | 7 | 5.5 | | |
| **Marital Status** | | | | 69.40 | <0.001 |
| Never married | 2,091 | 158 | 7.0 | | |
| Legally Married | 1,058 | 150 | 12.4 | | |
| Cohabiting | 1,414 | 254 | 15.2 | | |
| **Internet exposure** | | | | 0.63 | 0.428 |
| No | 3,827 | 464 | 10.8 | | |
| Yes | 736 | 98 | 11.8 | | |
| **Wealth index** | | | | 3.63 | 0.458 |
| Poorest | 985 | 104 | 9.6 | | |
| Poor | 937 | 112 | 10.7 | | |
| Middle | 791 | 106 | 11.8 | | |
| Rich | 809 | 107 | 11.7 | | |
| Richest | 1,041 | 133 | 11.3 | | |
| **Education level** | | | | 18.98 | <0.001 |
| No education | 202 | 28 | 12.2 | | |
| Primary | 2,683 | 349 | 11.5 | | |
| Secondary | 1,493 | 172 | 10.3 | | |
| Tertiary | 185 | 13 | 6.6 | | |
| **Residence** | | | | 2.48 | 0.116 |
| Urban | 1,505 | 204 | 11.9 | | |
| Rural | 3,058 | 358 | 10.5 | | |
| **Knowledge of Contraceptive** | | | | 2.37 | 0.123 |
| Yes | 4,176 | 525 | 11.2 | | |
| No | 387 | 37 | 8.7 | | |
| **Working status** | | | | 2.24 | 0.134 |
| Yes | 2,280 | 262 | 10.3 | | |
| No | 2,283 | 300 | 11.6 | | |
| **Region** | | | | 1.39 | 0.709 |
| Central | 978 | 128 | 11.6 | | |
| Eastern | 1,355 | 167 | 11.0 | | |
| Northern | 1,197 | 151 | 11.2 | | |
| Western | 1,033 | 116 | 10.1 | | |

In contrast, the ICC for the household random intercept model demonstrated that less than 1% of the total variation in abortion outcomes was attributable to differences between households, suggesting a minimal influence of household-level factors on abortion outcomes. This finding indicates that nearly all variation in abortion rates among adolescents occurs within households, i.e., at individual level, with only a slightly notable variation observed between regions. Furthermore, the significant likelihood ratio test p-value (0.011) supports the application of multilevel modelling at the regional level of

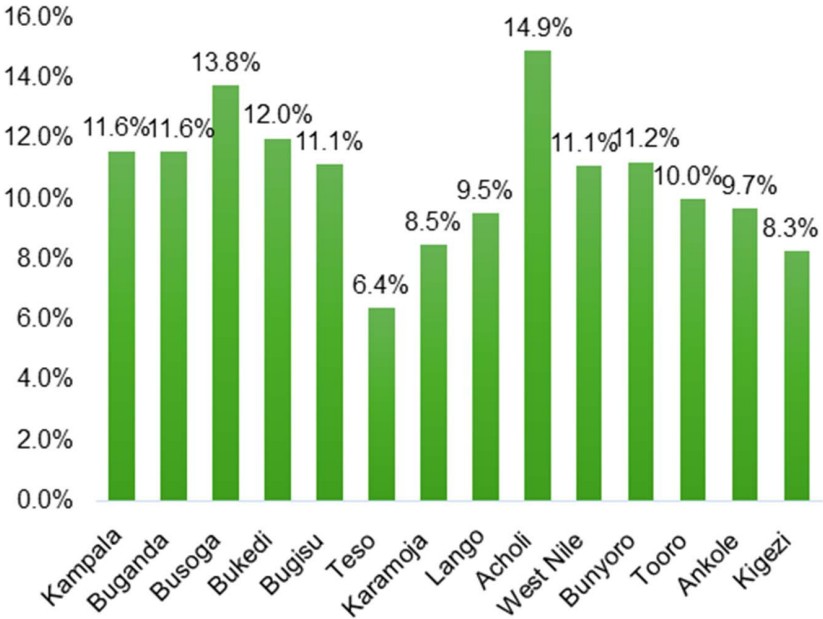

**Fig 1. Prevalence of adolescent abortion across sub-regions in Uganda.**

**Table 2. Assessing for variation in adolescent abortion at cluster level.**

| Measure of variation | Household Random intercept model | Regional Random intercept Model |
|---|---|---|
| Variance | 0.012 (0.004 - 0.282) | 0.168 (0.064 – 0.439) |
| ICC | 0.006 (0.002 - 0.092) | 0.049 (0.019 – 0.118) |
| Likelihood ratio test p-value | 0.424 | 0.011 |

clustering. Notably, although adolescents with no education had the highest odds of unsafe abortion overall, the magnitude of this association varied slightly across regions with the regional likelihood ratio of 0.011, implying that contextual regional factors may modify the strength of the education–abortion relationship among adolescents.

Fig 2 shows that the majority of variation in adolescent abortions occurs within regions and households rather than between them. Only 4.9% of the total variation is attributable to regional differences while 95.1% occurs within regions.

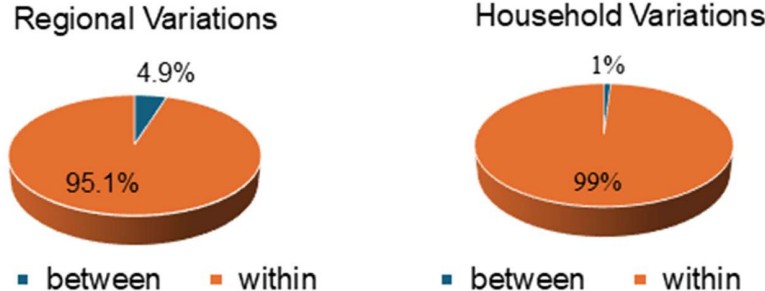

**Fig 2. Proportion of variation in adolescent abortion at cluster levels.**

Similarly, household-level differences account for just 1% of the variation with 99% occurring within households indicating that adolescent abortion rates are driven more by individual level factors than by regional or household clustering.

### Factors associated with adolescent abortion in Uganda

Table 3 presents the results from the multivariable logistic regression model analyzing abortion among adolescents while accounting for regional-level correlations with individual-level characteristics nested within regions. Additionally, only variables with a p-value < 0.10 in the bivariable analysis were included in the multivariable model.

The findings show that adolescents who had their first sexual intercourse before the age of 15 years had 3.44 times higher odds of abortion (AOR = 3.44, 95% CI: 1.55- 7.60) and those who initiated sexual activity between 15–17 years had 2.24 times higher odds (AOR = 2.24, 95% CI: 1.03 - 4.89) compared to adolescents whose first sexual encounter occurred after the age of 20.

Marital status was significantly associated with abortion outcomes. The odds of abortion were 2.01 times higher (AOR = 2.01, 95% CI: 1.58 - 2.56) among legally married adolescents and 2.44 times higher (AOR = 2.44, 95% CI: 1.97 - 3.02) among cohabiting adolescents, compared to those who were never married.

Education level also influenced abortion rates. Adolescents with primary education had nearly three times higher odds of abortion (AOR = 2.99, 95% CI: 1.60–5.60), and those with secondary education had similar odds (AOR = 3.01, 95% CI: 1.63–5.55) compared to adolescents with tertiary or higher education. However, adolescents with no education had almost six times the odds of abortion (AOR = 5.80, 95% CI: 1.90–7.36).

## Discussion

The research presented here used the 2022 Uganda Demographic and Health Survey (UDHS) to study the various determinants influencing adolescent abortions in Uganda. Descriptive statistics and mixed-effects logistic regression was used to determine factors associated with adolescent abortion, while accounting for regional effects. Factors that significantly influence adolescent abortions were age at first sex, marital status and education level.

The study found that 4.9% of variation in abortions is attributable to differences in regions implying that each region has some unique characteristics that predispose these adolescents to having abortions. Uganda's regions differ greatly

**Table 3. Multivariable analysis of factors associated with adolescent abortion.**

| Factor | AOR (95% CI) | p-value |
|---|---|---|
| **Variable** | | |
| **Age at first sex** | | |
| < 15 | 3.44 (1.55- 7.60) | 0.002 |
| 15-17 | 2.24 (1.03 - 4.89) | 0.043 |
| 18-20 | 1.32 (0.59 - 2.93) | 0.503 |
| > 20 | Ref | Ref |
| **Marital status** | | |
| Never married | Ref | Ref |
| Legally Married | 2.01 (1.58 - 2.56) | <0.001 |
| Cohabiting | 2.44 (1.97 - 3.02) | <0.001 |
| **Education level** | | |
| No education | 5.8 (1.90 - 7.36) | <0.001 |
| Primary | 2.99 (1.60 - 5.60) | 0.001 |
| Secondary | 3.01 (1.63 - 5.55) | <0.001 |
| Tertiary | Ref | Ref |

in socio-economic development, access to health services, cultural norms and population dynamics [25,26]. For instance, the Central region, which is more urbanized, has higher levels of education and better access to health facilities, but also higher rates of premarital sexual activity and exposure to factors known to increase risky sexual behaviors, such as peer influence and greater anonymity [27,28]. On the other hand, the Northern region has historically experienced greater conflict and higher poverty levels and faces persistent gaps in access to contraceptives and youth friendly services, potentially limiting adolescents' ability to prevent unintended pregnancies [29]. The Eastern region is characterized by early marriage practices in some communities, limited sexual health information and high fertility norms [30]. Finally, despite higher levels of health service availability, parts of the Western region still have strong cultural restrictions around adolescent sexuality, which may hinder open discussion and access to reproductive health services [31]. Differences in socio-cultural norms, service availability and economic conditions create varying risk settings across Uganda, which support the observed variation among regions. These findings align with existing global research, emphasizing the complex interplay of individual factors and region-level factors in shaping adolescent reproductive health outcomes [32–34].

Age at first sex plays a significant role in adolescent abortions. Adolescents who engage in sexual activities before the age of 15 are at a significantly higher risk of unintended pregnancies and abortion compared to those who had sex first above 20 years. This increased risk maybe due to a lack of sexual education and preparedness among these adolescents as stated by Sanni and Namukasa [35,36]. In the Ugandan context, early sexual debut is frequently linked to coercion and sexual violence and it remains a significant concern [37]. According to the Uganda Demographic and Health Survey (UDHS) and reports from the Uganda Police Annual Crime Reports [38,39], a substantial proportion of girls experience sexual violence, defilement, or coerced sex [38]. For example, the UDHS consistently shows that about 1 in 5 women aged 15–19 report having experienced sexual violence and defilement remains one of the most commonly reported crimes against children in Uganda, with thousands of cases recorded annually [39,40]. Such coercive experiences are strongly associated with unprotected sex, low contraceptive use and unintended pregnancies [37]. Additionally, socio-economic disadvantages and psychological factors such as lower self-esteem and higher pregnancy susceptibility among peer pressure may contribute to increased abortions and pregnancies among this group [41]. Furthermore, younger adolescents are often more vulnerable to coercion and sexual violence, which have been shown to further increase the likelihood of unprotected sex and unintended pregnancies as well as abortions [36,42].

Furthermore, this study found that legally married and cohabiting adolescents are more likely to experience unintended pregnancies and seek abortions compared to those who never married. This could be due to increased sexual activity within marital groups and cohabiting relationships who are not ready for childbearing with little use of family planning within these unions [43,44].

This study also found that education level significantly influences odds of ever having had an abortion, which is consistent with several studies [45–47] that found higher educational attainment is associated with better reproductive health outcomes, including lower abortion rates, due to improved access to information and resources. Another study also noted that adolescents with higher education levels are more likely to use contraceptives effectively and have lower rates of unintended pregnancies [48]. Conversely, adolescents with lower educational attainment may lack access to reproductive health services and information, which can lead to higher abortion rates [49]. A study by Kost and Henshaw [50] highlighted the importance of comprehensive sexual education in reducing abortion rates by promoting effective contraceptive use among adolescents.

Assessing variation at different hierarchical levels, the study found a minimal influence at the household level, suggesting that decisions related to abortion may not be as strongly tied to family characteristics as there maybe broader social, economic and cultural factors within communities [51,52]. A study on the incidence of Induced Abortion in Uganda found that regional estimates for unwanted pregnancies differ greatly with West Nile, Eastern, East Central, and North have the highest rates between 164 and 198 per 1,000 women, whereas Karamoja has the lowest incidence of 61 per 1,000 women [13]. Another study found out that women who lived in regions outside Kampala, Uganda's capital, were less likely

to utilize maternal health services and family planning resources. Further, these women had poor access to adolescent friendly services and limited access to health education [53]. In the Eastern and Northen Uganda, many believe that once a girl reaches puberty and begins menstruating, she is ready for marriage. Early marriage is common practice and considered honorable to parents but this reduces girl's ability to make informed decisions about their health [54,55]. Cultural practice of early marriage is just one possible explanation of how reproductive health decisions, particularly among adolescents, are often influenced by social norms external to the household, in addition to community-level access to healthcare services [52,56,57].

The noticeable variation at the region level highlights the importance of region and societal influences. This aligns with previous studies that have indicated contextual factors such as access to healthcare facilities, the availability of reproductive health education, cultural norms religious beliefs and local policies regarding abortion all contribute significantly to adolescent reproductive health decisions [58–60]. For example, regions where healthcare facilities are more accessible have higher rates of abortion among adolescents. Regions with strong religious or cultural opposition to abortion may see lower rates [61,62].

## Conclusion

This study found that variations in abortion rates among adolescents in Uganda could be partly explained by regional differences, indicating that each region has unique factors influencing abortion practices. By including regional random effects in our analysis, we accounted for unobserved region-level variability even though no specific region-level variables were identified as significant. The use of regional random effects is a key contribution of this study and captures the influence of unobserved region-specific factors.

## Limitations

No specific regional-level variables were able to be identified as significant in this study, despite the finding that regional differences exist. This highlights the need for further research to explore the regional factors that significantly contribute to the differences in abortion rates among adolescents in Uganda.

Another important point is the lack of information on the reasons for abortion within the Uganda Demographic and Health Survey (UDHS) dataset. Understanding the motivations behind abortion decisions among adolescents would provide a more comprehensive understanding on the factors driving these choices. Unfortunately, the UDHS does not collect data on these motivations, which limits our ability to fully understand the context in which these decisions are made. Additionally, the 2022 UDHS does not collect information on gender-based violence nor domestic violence for those women who have never married or cohabitated with a partner. The lack of data for these issues limits understanding and analysis of the relationship between these factors and unintended pregnancy particularly for young women.

Lastly, the study relies on self-reported data which is subject to reporting bias, especially for sensitive topics such as abortion. Adolescents may underreport or misreport their abortion experiences due to stigma or fear of judgment which can possibly lead to underestimation of the true prevalence of abortion among this population.

## Ethics declaration

This study was conducted as a secondary analysis using available survey data from The Demographic and Health Survey Program and requested from the Uganda Bureau of Statistics. Therefore, this study was not subject to its own ethical review since it was analysis of deidentified publicly available data. All DHS surveys are approved by the ICF Institutional Review Board, as well as an Institutional Review Board in the host country. This ensures all protocols for data collection follow regulations for the protection of human subjects and that all subjects provide consent to participate.

## Acknowledgments

We are grateful to researchers and authors who made their manuscripts and reports open access that enabled us access their papers during the development of literature. we are also grateful to our affiliated institutions for offering a conducive environment that enabled us to write this paper. Lastly our gratitude goes to Uganda Bureau of Statistics (UBOS) that provided us with the UDHS 2022 dataset that was used in this study.

## Author contributions

**Conceptualization:** Stephen Mungau, Joan Nanteza, Genevieve Dupuis.

**Formal analysis:** Stephen Mungau.

**Methodology:** Stephen Mungau, Genevieve Dupuis.

**Supervision:** Genevieve Dupuis.

**Writing – original draft:** Stephen Mungau.

**Writing – review & editing:** Stephen Mungau, Joan Nanteza, Genevieve Dupuis.

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
