## [Decision Letter · Decision Letter 0]

24 Oct 2025

PGPH-D-25-01751

Multilevel analysis of factors associated with abortion among adolescents in Uganda insights from UDHS 2022 Dataset..

Dear author.

Thank you for submitting your manuscript to PLOS Global Public Health. After careful consideration, we feel that it has merit but does not fully meet PLOS Global Public Health’s publication criteria as it currently stands. Therefore, we invite you to submit a revised version of the manuscript that addresses the points raised during the review process.

We look forward to receiving your revised manuscript.

Kind regards,

Niguss Cherie Bekele, PhD

Academic Editor

Journal Requirements:

1.  Please send a completed 'Competing Interests' statement, including any COIs declared by your co-authors. If you have no competing interests to declare, please state "The authors have declared that no competing interests exist". Otherwise please declare all competing interests beginning with the statement "I have read the journal's policy and the authors of this manuscript have the following competing interests:"

2. In the online submission form, you indicated that The dataset can be accessible by request from UBOS and do-file/analysis codes for this study are available from the corresponding author upon request.

3. Uploaded as supplementary information.

Additional Editor Comments (if provided):

Comments to the Author

1. Does this manuscript meet PLOS Global Public Health’s publication criteria? Is the manuscript technically sound, and do the data support the conclusions? The manuscript must describe methodologically and ethically rigorous research with conclusions that are appropriately drawn based on the data presented. Yes

2. Has the statistical analysis been performed appropriately and rigorously? Yes

3. Have the authors made all data underlying the findings in their manuscript fully available (please refer to the Data Availability Statement at the start of the manuscript PDF file)?

The PLOS Data policy requires authors to make all data underlying the findings described in their manuscript fully available without restriction, with rare exception. The data should be provided as part of the manuscript or its supporting information, or deposited to a public repository. For example, in addition to summary statistics, the data points behind means, medians and variance measures should be available. If there are restrictions on publicly sharing data—e.g. participant privacy or use of data from a third party—those must be specified. Yes

4. Is the manuscript presented in an intelligible fashion and written in standard English?

PLOS Global Public Health does not copyedit accepted manuscripts, so the language in submitted articles must be clear, correct, and unambiguous. Any typographical or grammatical errors should be corrected at revision, so please note any specific errors here. Yes

5. Review Comments to the Author

Please use the space provided to explain your answers to the questions above. You may also include additional comments for the author, including concerns about dual publication, research ethics, or publication ethics. (Please upload your review as an attachment if it exceeds 20,000 characters) The manuscript seems reasonably written. However authors have been a bit scanty with the discussion ad interpretation of results. More work is needed in the discussion especially to bring meaning.

Gender based violence variables have also not been considered in the analysis, a limitation which must be acknowledged, and explained why

6. PLOS authors have the option to publish the peer review history of their article (what does this mean?). If published, this will include your full peer review and any attached files.

Do you want your identity to be public for this peer review? For information about this choice, including consent withdrawal, please see our Privacy Policy. No

Confidential to Editor

1. Do you have any potential or perceived competing interests that may influence your review? Please review our Competing Interests policy and declare any potential interests. If you have no competing interests, please write "I have no competing interests." None

2. Did you receive any assistance in preparing this review (e.g. from a post-doc or graduate student)? If yes, please include their name below. (optional) NO

3. If accepted, do you think this submission should be highlighted on the journal website and/or to the media? (optional) Yes, on the journal website only

Do you want to get recognition for this review on a Web of Science researcher profile?

If you opt in, your Web of Science profile will automatically be updated to show a verified record of this review in full compliance with the journal’s review policy. If you don’t have a Web of Science profile, you will be prompted to create a free account.

Yes

Reviewers' comments:

Reviewer's Responses to Questions

**Comments to the Author**

1. Does this manuscript meet PLOS Global Public Health’s publication criteria?

Reviewer #1: Yes

2. Has the statistical analysis been performed appropriately and rigorously?

Reviewer #1: Yes

3. Have the authors made all data underlying the findings in their manuscript fully available (please refer to the Data Availability Statement at the start of the manuscript PDF file)?

Reviewer #1: Yes

4. Is the manuscript presented in an intelligible fashion and written in standard English?

Reviewer #1: Yes

Reviewer #1: The manuscript seems reasonably written. However authors have been a bit scanty with the discussion ad interpretation of results. More work is needed in the discussion especially to bring meaning.

Gender based violence variables have also not been considered in the analysis, a limitation which must be acknowledged, and explained why

**Do you want your identity to be public for this peer review?** For information about this choice, including consent withdrawal, please see our Privacy Policy

Reviewer #1: No

---

## [Editor Report · Decision Letter 1]

28 Dec 2025

Multilevel analysis of factors associated with abortion among adolescents in Uganda insights from UDHS 2022 Dataset.

PGPH-D-25-01751R1

Dear author

We are pleased to inform you that your manuscript 'Multilevel analysis of factors associated with abortion among adolescents in Uganda insights from UDHS 2022 Dataset.' has been provisionally accepted for publication in PLOS Global Public Health.

IMPORTANT: The editorial review process is now complete. PLOS will only permit corrections to spelling, formatting or significant scientific errors from this point on wards. Requests for major changes, or any which affect the scientific understanding of your work, will cause delays to the publication date of your manuscript.

Best regards,

Niguss Cherie Bekele, PhD

Academic Editor